# Current Practices on Diagnosis, Prevention and Treatment of Post-Transplant Lymphoproliferative Disorder in Pediatric Patients after Solid Organ Transplantation: Results of ERN TransplantChild Healthcare Working Group Survey

**DOI:** 10.3390/children8080661

**Published:** 2021-07-29

**Authors:** Alastair Baker, Esteban Frauca Remacha, Juan Torres Canizales, Luz Yadira Bravo-Gallego, Emer Fitzpatrick, Angel Alonso Melgar, Gema Muñoz Bartolo, Luis Garcia Guereta, Esther Ramos Boluda, Yasmina Mozo, Dorota Broniszczak, Wioletta Jarmużek, Piotr Kalicinski, Britta Maecker-Kolhoff, Julia Carlens, Ulrich Baumann, Charlotte Roy, Christophe Chardot, Elisa Benetti, Mara Cananzi, Elisabetta Calore, Luca Dello Strologo, Manila Candusso, Maria Francelina Lopes, Manuel João Brito, Cristina Gonçalves, Carmen Do Carmo, Xavier Stephenne, Lars Wennberg, Rosário Stone, Jelena Rascon, Caroline Lindemans, Dominik Turkiewicz, Eugenia Giraldi, Emanuele Nicastro, Lorenzo D’Antiga, Oanez Ackermann, Paloma Jara Vega

**Affiliations:** 1Paediatric Liver, Gastrointestinal and Nutrition Centre, School of Medicine, King’s College Hospital, King’s College London, Denmark Hill, London SE5 9RS, UK; alastair.baker@nhs.net (A.B.); emer.fitzpatrick@kcl.ac.uk (E.F.); 2Servicio de Hepatología Pediátrica, Hospital Universitario La Paz, 28046 Madrid, Spain; esteban.frauca@salud.madrid.org (E.F.R.); gmbartolo@salud.madrid.org (G.M.B.); paloma.jara@transplantchild.eu (P.J.V.); 3Lymphocyte Pathophysiology in Immunodeficiencies Group, La Paz Institute of Biomedical Research (IdiPAZ), Hospital Universitario La Paz and Center for Biomedical Network Research on Rare Diseases (CIBERER U767), 28046 Madrid, Spain; jmtorres@clinic.cat; 4Servicio de Nefrología Pediátrica, Hospital Universitario La Paz, 28046 Madrid, Spain; aamelgar@salud.madrid.org; 5Servicio de Cardiología Pediátrica, Hospital Universitario La Paz, 28046 Madrid, Spain; lggueretasilva@salud.madrid.org; 6Pediatric Gastroenterology Intestinal Rehabilitation Unit, University Hospital La Paz, 28046 Madrid, Spain; erboluda@salud.madrid.org; 7Pediatric Hemato-Oncology Department, Hospital Universitario La Paz, 28046 Madrid, Spain; yasmina.mozo@salud.madrid.org; 8Department of Pediatric Surgery and Organ Transplantation, Children’s Memorial Health Institute, 04-730 Warsaw, Poland; d.broniszczak@ipczd.pl (D.B.); p.kalicinski@ipczd.pl (P.K.); 9Department of Nephrology and Kidney Transplantation, Children’s Memorial Health Institute, 04-730 Warsaw, Poland; w.jarmuzek@ipczd.pl; 10Department of Pediatric Hematology and Oncology, Hannover Medical School, 30625 Hannover, Germany; maecker-kolhoff.britta@mh-hannover.de; 11Clinic for Paediatric Pneumology, Allergology, and Neonatology, Hannover Medical School, 30625 Hannover, Germany; carlens.julia@mh-hannover.de; 12Division of Paediatric Gastroenterology and Hepatology, Children’s Hospital, Hannover Medical School, 30625 Hannover, Germany; baumann.u@mh-hannover.de; 13Service de Pneumologie Pédiatrique, Hôpital Necker-Enfants Malades, AP-HP, Université Paris, 75015 Paris, France; charlotte.roy@aphp.fr; 14Service de Chirurgie Pédiatrique, Hôpital Necker-Enfants Malades, AP-HP, Université Paris Descartes, 75015 Paris, France; christophe.chardot@aphp.fr; 15Pediatric Nephrology, Dialysis and Transplant Unit, Department of Women’s and Children’s Health, Azienda Ospedaliera di Padova, 35128 Padova, Italy; elisa.benetti@aopd.veneto.it; 16Unit of Paediatric Gastroenterology, Digestive Endoscopy, Hepatology and Care of the Child with Liver Transplantation, Department of Women’s and Children’s Health, Azienda Ospedaliera di Padova, 35128 Padova, Italy; mara.cananzi@aopd.veneto.it; 17Unit of Paediatric Onco-Haematology, Department of Women’s and Children’s Health, Azienda Ospedaliera di Padova, 35128 Padova, Italy; elisabetta.calore@unipd.it; 18Nephrology Unit, Bambino Gesù Children’s Research Hospital, IRCCS, 00165 Rome, Italy; luca.dellostrologo@opbg.net; 19Department of Hepatology and Gastroenterology, Bambino Gesù Children Hospital, 00165 Rome, Italy; manila.candusso@opbg.net; 20Department of Paediatric Surgery, Centro Hospitalar e Universitário de Coimbra, and Faculty of Medicine, University of Coimbra, 3000-075 Coimbra, Portugal; mfrancelina@yahoo.com; 21Department of Paediatric Oncology and Centro de Investigação e Formação Clínica, Hospital Pediátrico, Centro Hospitalar e Universitário de Coimbra, 3000-075 Coimbra, Portugal; mjbrito@chuc.min-saude.pt; 22Paediatric Liver Transplant Unit, Centro Hospitalar e Universitário de Coimbra, 3000-075 Coimbra, Portugal; cristina.campos.goncalves@gmail.com; 23Paediatric Nephrology Unit, Hospital Pediátrico, Centro Hospitalar e Universitário de Coimbra, 3000-075 Coimbra, Portugal; pmccferreira@gmail.com; 24Laboratoire d’Hépatologie Pédiatrique et Thérapie Cellulaire, Unité PEDI, Institut de Recherche Expérimentale et Clinique, Université Catholique de Louvain (UCLouvain), 1200 Brussels, Belgium; xavier.stephenne@uclouvain.be; 25Department of Transplantation Surgery, Karolinska University Hospital, 171 76 Stockholm, Sweden; lars.wennberg@sll.se; 26Unidade de Nefrologia e Transplantação Renal, Serviço de Pediatria Médica, Departamento de Pediatria, Hospital de Santa Maria, Centro Académico de Medicina, Universidade de Lisboa, 1649-028 Lisboa, Portugal; rstone@chln.min-saude.pt; 27Centre for Paediatric Oncology and Haematology, Vilnius University Hospital Santaros Klinikos, 08406 Vilnius, Lithuania; jelena.rascon@gmail.com; 28Princess Maxima Center for Pediatric Oncology, Pediatric Blood and Marrow Transplantation Program, University Medical Center Utrecht, Utrecht University, 3584 CS Utrecht, The Netherlands; c.a.lindemans@prinsesmaximacentrum.nl; 29Department of Pediatrics, Skåne University Hospital, 222 42 Lund, Sweden; dominik.turkiewicz@gmail.com; 30Pediatric Oncology, Hospital Papa Giovanni XXIII, 24127 Bergamo, Italy; egiraldi0@gmail.com; 31Pediatric Hepatology, Gastroenterology and Transplantation, Hospital Papa Giovanni XXIII, 24127 Bergamo, Italy; enicastro@asst-pg23.it (E.N.); ldantiga@hpg23.it (L.D.); 32Pediatric Hepatology, National Centre for Biliary Atresia, Université París-Saclay, APHP, Hôpital Bicêtre, 94270 Le Kremlin-Bicêtre, France; oanez.ackermann@bct.aphp.fr; 33La Paz Institute of Biomedical Research, IdiPAZ, Hospital Universitario La Paz, 28046 Madrid, Spain

**Keywords:** PTLD, post-transplant lymphoproliferative disorder, pediatric, solid organ transplantation, immunosuppression, Epstein–Barr virus

## Abstract

(1) Background: Post-transplant lymphoproliferative disease (PTLD) is a significant complication of solid organ transplantation (SOT). However, there is lack of consensus in PTLD management. Our aim was to establish a present benchmark for comparison between international centers and between various organ transplant systems and modalities; (2) Methods: A cross-sectional questionnaire of relevant PTLD practices in pediatric transplantation was sent to multidisciplinary teams from 17 European center members of ERN TransplantChild to evaluate the centers’ approach strategies for diagnosis and treatment and how current practices impact a cross-sectional series of PTLD cases; (3) Results: A total of 34 SOT programs from 13 European centers participated. The decision to start preemptive treatment and its guidance was based on both EBV viremia monitoring plus additional laboratory methods and clinical assessment (61%). Among treatment modalities the most common initial practice at diagnosis was to reduce the immunosuppression (61%). A total of 126 PTLD cases were reported during the period 2012–2016. According to their histopathological classification, monomorphic lesions were the most frequent (46%). Graft rejection after PTLD remission was 33%. Of the total cases diagnosed with PTLD, 88% survived; (4) Conclusions: There is still no consensus on prevention and treatment of PTLD, which implies the need to generate evidence. This might successively allow the development of clinical guidelines.

## 1. Introduction

Post-transplant lymphoproliferative disease (PTLD) is a significant complication of solid organ transplantation (SOT), and it is the most common post-transplant malignancy in children. PTLD is largely a disease of the modern medical era, directly linked to the use of increasingly potent immunosuppressive regimens [1,2]. The risk factor profile and pathogenesis of PTLD are not completely defined. Although Epstein–Barr virus (EBV) is associated with PTLD, not all patients with high viral load develop this malignancy, and some PTLD tumors are EBV-negative. These apparent contradictions make it difficult to predict which patients will develop PTLD, of what severity or site and how they will respond to treatment [3,4].

Risk factors for SOT–PTLD include EBV recipient–donor seromismatch (recipient−/donor+) due to a lack of recipient preformed anti-EBV cytotoxic immunity, T cell immunosuppression with anti-thymocyte globulin or other immunosuppressive medications and the presence of active cytomegalovirus (CMV) infection, which can induce immune senescence and is a surrogate for poor cellular immunity [2,5,6,7]. A major risk factor specific to SOT is the use of intestinal or multivisceral transplants due to a high volume of donor lymphoid tissue contained in the graft, which is subject to expansion when exposed to EBV and in a highly immunosuppressed environment [8]. PTLD occurs in 2–15% of pediatric SOT patients depending on the organ transplanted and the immunosuppression used [2,6,8,9,10,11,12].

In SOT, most childhood PTLD occurs during the period of the most intensive T cell suppression to prevent graft rejection, usually within the first 2 years after transplantation. Late-onset PTLD can occur in SOT recipients due to lifelong immunosuppression, but it accounts for less than 10% of cases [13]. The highest incidence of PTLD occurs at primary EBV seroconversion due to de novo infection or when it is acquired from passenger lymphocytes in the graft. SOT donors are typically older, and therefore more frequently EBV positive, than their organ recipients. Due to the paucity of EBV-negative donors, matching of donor and recipient pairs by EBV status to reduce the risk of PTLD has proven to be difficult.

It is important to note that the consequences of PTLD development in terms of mortality and morbidity are related not only to the tumor proliferation itself, but also to the potential negative effects of its treatment, such as graft rejection after reducing immunosuppression or the adverse effects of various possible drugs whose use might be considered necessary.

From a pathological point of view, PTLD can vary from an infection-like appearance to a frank lymphoma. It can evolve progressively from EBV reactivation/infection to a polyclonal disorder and more aggressive monoclonal disease [4,14]. The cornerstone of PTLD treatment is to enable immune reconstitution in the host for an effective antitumor response, and antineoplastic chemotherapy and immunotherapy. Current guidelines recommend reducing immunosuppression whenever possible; however, the documented effectiveness varies between studies [11]. There are no published criteria predicting the response to reducing or stopping immunosuppression in pediatric patients.

Given the lack of consensus in PTLD management, we aimed to establish a present benchmark for comparison between international centers and between various organ transplant systems and modalities. Comparison between centers in their approaches to clinical management might be expected to reveal a baseline for research projects or to reveal the need to design clinical guidelines or protocols to standardize and improve care.

## 2. Materials and Methods

The Healthcare Working Group of the European Reference Network on Pediatric Transplantation (ERN TransplantChild) agreed on the design of a cross-sectional questionnaire of relevant PTLD practices in pediatric transplantation, for which no current consensus exists. This questionnaire was related to the clinical practice followed by physicians of multidisciplinary teams involved in transplantation and dedicated to diagnosis, prevention and treatment of PTLD cases.

The questionnaire was divided into 2 parts. The first part was about the centers’ approach to immunosuppression (IS), CMV prophylaxis, antiviral treatment and preemptive strategies. For example, combining serial quantitative EBV DNA monitoring in peripheral blood with interventions that might lower risk, triggered by EBV DNA levels predictive of PTLD development occurring before onset of clinical disease [5]; intravenous immunoglobulin treatment; EBV monitoring; biomarkers and imaging tests; and PTLD practices with respect to indications for rituximab, EBV-specific cytotoxic T–cell treatment (CTLs), chemotherapy or surgical removal.

The second part of the questionnaire focused on collecting all the PTLD cases diagnosed in the participating centers during the inclusive 5-year period of 2012–2016. Data about time of diagnosis, survival, type and location of lesions and the presence of rejection or relapse after remission were collected, with the objective of expressing the general impact that this complication has in pediatric SOT. The time of diagnosis was defined as the period between transplantation and PTLD diagnosis. Patients who presented with PTLD within the first 12 months post-transplantation were categorized as “early–onset”, recipients presenting with the disease beyond this time but less than 10 years after transplantation were considered as “late-onset” and patients with PTLD after the tenth post-transplant year were categorized as “very-late-onset” [15].

All available tissue samples at PTLD diagnosis were classified according to the 2017 World Health Organization (WHO) classification [16]. There are 4 pathologic categories: (1) nondestructive PTLD (plasmocytic, hyperplasia, infectious mononucleosis and florid follicular hyperplasia), (2) polymorphic PTLD, (3) monomorphic PTLD (B cell, T cell, and NK cell) and (4) classic Hodgkin lymphoma PTLD [16].

Pediatric SOT (heart, intestinal, multivisceral, kidney, liver, lung and pancreas) departments, as well as available hematologists, oncologists, pathologists and other experts from 17 European centre members of ERN TransplantChild were invited to complete the online survey (https://ec.europa.eu/eusurvey/runner/PTLD_Audit accessed on 1 June 2021) from 2018 to 2019. Since participation was optional, decision to participate to the survey was considered as (implicit) consent to participate. The study aims were explained to all candidate centers prior to performing the survey.

The descriptive statistics used to express the data include medians with IQR. A chi-squared test for trend was used to compare proportions between groups and a Mann–Whitney test used to compare continuous non-parametric variables. A *p*-value < 0.05 was considered significant. Analyses were performed by using R statistical software, v3.6.1 (R Foundation for Statistical Computing, Vienna, Austria).

## 3. Results

An invitation to participate in the survey was sent by e-mail to the representatives of 17 European SOT centers from 11 countries. A total of 34 SOT programs from 13 European centers and 9 countries participated in the study. All types of SOT programs were represented (11 liver, 9 kidney, 6 intestinal/multivisceral, 5 heart, 2 lung and 1 pancreas).

Of the 34 total SOT programs, 29 provided information about PTLD cases during the period 2012–2016: a total of 2329 transplants were performed and 181 new PTLD cases were diagnosed, (liver 115/1471 [7.8%]; kidney 33/656 [5%]; intestinal/multivisceral 15/69 [22%]; heart 14/77 [18%]; lung 4/56 [7.1%]; pancreas 0/0).

### 3.1. Centers’ PTLD Approach

#### 3.1.1. Immunosuppression and PTLD Prevention

The immunosuppression protocols for induction and maintenance varied according to the type of transplant. The induction treatment included the use of basiliximab or other biologics as inducing agents in 55% of cases. Maintenance of immunosuppression after transplantation was performed with 1 IS (22%), 2 ISs (19%) or 3 ISs (59%). The IS agents prescribed were steroids (93%), followed by tacrolimus (74%) and mycophenolate mofetil (59%).

Pretransplant EBV and CMV: donor/receptor EBV and CMV serology match/mismatch were considered in 12% and 62% of centers during donor selection, respectively.CMV prophylaxis and antiviral treatment: CMV prophylaxis was a common practice in SOT (SOT, 75%; kidney, 100%; intestinal/multivisceral, 80%; heart, 75%; liver, 64%; lung, 50%; pancreas, 0%). CMV serostatus was considered in the prophylaxis decision (55%), and the decision of prophylaxis was variable in each of the types of transplant, based on the perceived risk in each case (61%) versus universal prophylaxis (39%). Prophylaxis with ganciclovir/valganciclovir was common (81%), with a median duration of 3 months (IQR 3-6).Preemptive PTLD strategies: the decision to start preemptive treatment and its guidance was based on both EBV viremia monitoring plus additional laboratory methods and clinical assessment (61% of centers). Additional methods included detection and monitoring of virus T cell immunity by enzyme-linked immunospot (ELISPOT) (29%) or flow cytometry (9.6%) and IgM protein electrophoresis (16%). All programs reported performing EBV load surveillance in all SOT recipients, but with different frequencies. EBV DNAemia determinations by polymerase chain reaction were mainly performed in whole blood specimens (76%) and plasma specimens (20%). There were no specific criteria to define the EBV viremia cutoff value to start preemptive management. Within the preemptive strategies, the most common practice was a reduction in immunosuppression (86%), and occasionally the use of antivirals (15%) or rituximab (38%). The strategy for adjusting immunosuppression tacrolimus/cyclosporin target levels back was based on the decrease or clearance of EBV viremia (72%), whereas 12% of programs maintain tacrolimus/cyclosporin at low levels even when EBV viremia has decreased (“wait and see” strategy). Rituximab as a preemptive strategy in SOT varies considerably by type of transplant (SOT, 38%; liver, 50%; intestinal/multivisceral, 50%; kidney, 38%; heart, 25%; lung, 0%; pancreas, 0%).

#### 3.1.2. PTLD Diagnostic and Evaluation Tests

PTLD suspicion diagnosis was mainly based on imaging procedures and some biomarkers. Concerning the diagnostic imaging approach to PTLD, computed tomography (CT) was mainly used (94%), followed by positron-emission tomography (PET) (85%) and magnetic resonance (67%). Biomarkers were used in 63% (serum lactate dehydrogenase, 63%; fecal occult blood, 17%; other biomarkers, 38%). Invasive diagnostic measures such as allograft biopsy or lymphatic node biopsy, were generally performed in patients with clinical, imaging or endoscopic manifestations compatible with PTLD (100%), whenever technically feasible.

#### 3.1.3. PTLD Treatment

For the treatment of PTLD (Table 1), the most common initial practice at the diagnosis was to reduce the IS (61%), followed by reducing IS or stopping IS (30%) depending on each individual case, and stopping immunosuppression at the diagnosis (9.1%).

The indication for rituximab was based on various clinical and diagnostic measures (alone or in combination): individualized discussion (60%), histological diagnosis (pathologic category) monomorphic subtype (56%), histological detection of CD20+ B cells (44%), and EBV viral load plus clinical manifestations (36%). Rituximab was administered mostly once weekly to a total of four doses (70%). Treatment with cytotoxic T cell treatments was only considered for 22% of transplant programs, and its indication was based on refractoriness to initial treatment with persistence of EBV. Chemotherapy was considered in 79% of SOT programs, depending on the histological findings, as well as lack of response to rituximab (71% and 79%, respectively). Surgical removal was considered in 58% of transplant programs and depending on tumor surgical accessibility.

### 3.2. Clinical Outcomes of PTLD Cases

Twenty-two SOT programs provided specific information (e.g., demographics, diagnosis and PTLD histological type) of 126/181 clinical PTLD cases. A summary of clinical cases is presented in Table 2. The median time from transplantation to PTLD diagnosis was 12 months (IQR 6–42.2 months), with longer presentation time in kidney transplant patients and shorter time in lung transplant patients (lung, 1 month; liver, 12 months; intestinal/multivisceral, 12 months; heart, 36 months; kidney, 52 months).

According to time of post-transplant PTLD diagnosis, 49 cases were considered early–onset (39% of the total; mean time, 4.99 ± 3.09 months), 73 late-onset (58% of the total; mean time, 37.56 ± 26.72 months) and 4 very late onset (3.2% of the total; mean time, 131.2 ± 6.25 months).

EBV levels at PTLD diagnosis were lower in very late presentations (1 × 10^5^ ± 1.72 × 10^5^ copies/mL) compared with early (2.85 × 10^6^ ± 4.71 × 10^6^ copies/mL, *p* = 0.036) and late presentations (3.63 × 10^6^ ± 14.8 × 10^6^ copies/mL, *p* = 0.17).

The primary PTLD tumor locations are described in Table 3, with adenoids/tonsils and cervical lymph nodes being the most frequent localization (43%), followed by the gastrointestinal tract (33%). The transplanted organ was involved in 13% of PTLD cases.

According to their histopathological classification, monomorphic lesions were the most frequent (46%) followed by polymorphic (21%), early (19%) and Hodgkin lymphoma (14%). Monomorphic lesions had a different distribution according to time of PTLD diagnosis: early–onset cases (35%), late–onset cases (52%) and very–late-onset cases (75%).

Of the total 181 cases diagnosed with PTLD, 160 survived (all SOT programs, 88.%; liver, 92%; heart, 86%; kidney, 85%; lung, 75%; intestinal/multivisceral, 73%). Graft rejection after PTLD remission was higher among cardiac and intestinal/multivisceral cases (SOT, 33%; heart, 67%; intestinal/multivisceral, 46%; liver, 35%; kidney, 12%; lung, 0%).

## 4. Discussion

PTLD remains one of the most serious potential complications in transplanted children and the most frequent type of cancer in these patients. The incidence is significantly higher than in adult patients because 90% of these tumors are related to EBV infection; a high percentage of children are EBV-seronegative at the time of transplantation and therefore have a high risk of primary infection thereafter [17].

In recent years, a significant reduction in PTLD incidence in transplanted children has been reported as a consequence of the greater efficacy in its prevention [5,18]; however, it remains a disease associated with non-negligible morbidity and mortality rates in affected patients, for which, to date, there are no evidence-based clinical guidelines available for children with SOT.

Based on these premises, it was considered useful to design a survey whose results could reveal the degree of agreement between various pediatric SOT programs on aspects related to the prevention, diagnosis, and treatment of PTLD. Other surveys focused on the prevention or management of PTLD have previously been published. However, unlike the one presented here, they were focused on specific types of SOT or included both pediatric and adult patient transplantation programs [19,20].

The fact that 76% of the centers asked to participate in the survey responded, and 85% of the transplant programs active in those centers provided data on diagnosed PTLD cases, means participation was very high and reflects the benefits of being part of a reference network in pediatric transplantation, such as TransplantChild.

The current prevention of lymphoproliferative disease in transplanted children is mainly based on the identification of at risk patients and early treatment to prevent the development of PTLD [5,20,21]. The results of our survey found that all centers included still base the identification of those patients on the monitoring of EBV viral load in peripheral blood; however, progressively more centers (62% in our survey) incorporate adjunctive laboratory tests that could improve the specificity and positive predictive value of high viral load as a predictor of PTLD. Among the best studied and most promising are ELISPOT assays and tetramers measuring T–cell restoration or EBV-specific T–cell responses [22]. However, only 39% of the surveyed centers confirmed their use, indicating that we are still far from its widespread implementation. Randomized controlled trials on biomarkers are needed to provide sufficient evidence on aspects such as the frequency or type of biological sample to quantify viral load, as well as to determine a cutoff point for starting early treatment [23]. According to our results, these are all important aspects that remain undefined; thus, prospective studies employing the WHO international standardization assay for EBV quantification are needed to define a viral load threshold consensus, either in absolute value or rate of increase, at which early treatment should be initiated in different clinical risk settings [22,23,24,25]. As a complementary effort, the Research Working Group of the ERN TransplantChild is conducting a systemic review of other biomarkers for viral status or immune competence, in discovery or validation stages, which could be further explored on collaborative efforts within the network.

Although the decision of when to start treatment early should not be based only on an arbitrary viral load threshold but also on the individual risk factors detected and the rate viral load increase [26], the establishment of this threshold could be a useful tool to standardize prevention strategies. The high rate of variability reported between laboratories in the quantification of viremia currently represents an added difficulty for defining an effective cutoff value [27].

Once the patient risk has been defined, a large majority of centers base preemptive treatment on the reduction of immunosuppression (86%), with the aim of restoring the EBV-specific T–cell-mediated immunity and whose efficacy has been reported in pediatric patients with SOT [28]. Only 15% of centers use antivirals in this phase. The use of rituximab, which has been recommended as a second-line treatment in patients refractory to IS reduction, is still rare in the surveyed centers [29,30].

Other treatment options, mainly in hematopoietic stem cell transplantation, such as the infusion of cytotoxic T cell treatments, have marginal use in pediatric SOT according to the obtained results and probably related to the limited experience available and the drawbacks associated with this treatment modality [31].

In our survey, the centers mostly use the reduction of viremia to return to the previous levels of IS (72%), whereas only 12% maintain a low IS with close monitoring of rejection signs to increase it. A reduction in IS implies a risk of rejection for patients; thus, it should ideally be guided in its duration and intensity by some direct or indirect immunological marker, which is currently lacking in clinical practice and would allow this risk to be minimized.

The lack of evidence to support the use of antivirals in patients with a high viral load, beyond their efficacy reported in some uncontrolled trials in short series of patients, probably explains the strong agreement on their nonuse by 88% of the surveyed centers [32,33]. Given that the antiviral activity of these drugs is focused only on the lithic phase of EBV infection, its role in the development of PTLD should be verified to justify its use as an anticipated treatment in these patients.

Given the risks and costs of early treatments, such as rejection in the event of reducing IS or the potential adverse effects of rituximab or antivirals, the need for a better definition of patients at real risk of PTLD to design an individualized and aggressive treatment is a key aspect and is the only way to further improve prevention and reduce the incidence of PTLD.

All the surveyed centers use the WHO classification for staging the lesions, which represents a positive advance and a starting point towards a standardization of treatment protocols according to the histological type of PTLD.

Once a clinical suspicion has been raised, most of the surveyed centers reproduce the same diagnostic process with the use of imaging techniques, including the most recent ones, such as PET–CT, to establish a suspected diagnosis that will need to be confirmed with histological study of the lesions detected if biopsy is feasible.

Due to the heterogenic nature of PTLD, the therapeutic approach needs to consider the histology and clinical settings in which PTLD has developed [34]. Reduced immunosuppression (RI) or discontinuation of the IS is recommended earlier for all cases, and this was a universal practice in the centers included in our study. If RI is not enough, it is advisable to consider therapy with rituximab or EBV-CTL [35,36,37,38], if available. The use of rituximab in our cohort differs depending on histological, clinical and EBV detection findings. Rituximab should be considered as monotherapy for the next level of treatment in CD20+ PTLD in patients with progressive disease after RI. Frontline rituximab monotherapy in nondestructive or polymorphic lesions is associated with response rates of approximately 60–65% and up to 80%, a benefit that is offset by approximately 20–30% treatment failures and 2-year overall survival of only about 50% [18,39,40]. Accordingly, rituximab often needs to be combined with other therapies, such as RI if feasible, or EBV-CTLs.

A low-dose chemotherapy regimen is recommended for EBV associated with graft rejection, fulminant PTLD symptoms or when a high disease burden occurs. If low-dose chemotherapy fails to achieve an adequate response, or if the histology shows NK/T cells, Hodgkin’s lymphoma, Burkitt’s lymphoma, or EBV-negative PTLD, the use of conventional-dose chemotherapy is recommended. Although the multiagent chemotherapy regimen of cyclophosphamide, hydroxydaunorubicin, oncovin and prednisone can produce durable responses, it has been associated with high treatment-related mortality of up to 30% for PTLD [41,42,43,44]. Most centers in our study use chemotherapy either in histologically more aggressive cases or in those with a noncomplete response to rituximab.

Regarding the treatment of diagnosed PTLD, the results of our survey indicate that there is strong agreement among centers as to reduction/cessation of IS or chemotherapy. On the other hand, the indications and role in the PTLD treatment protocol of rituximab or the EBV-CTLs still need to be better defined according to the various histological types and the prognosis of each case.

There are no established treatment regimens for relapsed or refractory PTLD in SOT [40,45]. In the rare event that PTLD is localized and not responsive to RI or rituximab, local treatment measures such as surgery or radiotherapy can be considered. Complete surgical resection is generally sought when there is a low chance for surgery-related morbidity, or when debulking is thought beneficial for rapid improvement of severe or life-threatening complications from a developing lymphomatous mass [46]. Radiotherapy and more intensive chemotherapy with anthracyclines and other agents are not considered front-line therapy for EBV-positive B-cell PTLD, and these agents are restricted to patients with refractory or recurrent disease, T–cell lymphomas or Hodgkin-like PTLD or central nervous system PTLD [1].

The prognosis of PTLD is related to the morphological subtype, the presence of EBV, timing after transplant and patient-specific factors, such as comorbidities and suitability for intensive therapy. Nondestructive PTLD within 1 year of SOT is associated with a relatively good prognosis, and approximately 80% are cured with standard sequential chemo-immunotherapy, whereas late-onset PTLD augurs a poorer prognosis [6].

In relation to the PTLD cases reported by the centers surveyed, the main consideration is that this is a very large series of PTLD cases, considering its incidence has been estimated 10% in most published series and that this number was only been possible due to the multicenter nature of the study. We cannot estimate a PTLD prevalence because we do not have the number of patients followed up in the various units during the 5-year period in which the 126 cases reported by the 29 transplant programs were diagnosed.

The beginnings of PTLD typically occur in the first year after transplant (39%), as previous series have reported [47]. Although the possibility of developing PTLD persists in the long term, very late onset cases are infrequent (3.2% of all cases). Thus, PTLD surveillance should be maintained in the long term, although probably in a less close way than during the first post-transplantation years. The difference in the onset of PTLD presentation between the various transplanted organs could be related to the different global intensity of IS prescribed for each type of transplant.

In relation to the primary location of tumors, our results confirm that these can develop in any body area; however, as already reported in other series, the ear, nose, throat/neck area and gastrointestinal location are the most frequent for the development of PTLD in children with SOT (75% of the total cases reported) [17]. This great diversity in its location implies that a high degree of diagnostic suspicion should be maintained for any new lesion regardless of its location.

Likewise, the distribution of cases according to the histopathological type confirms that PTLD can be variable, with a minority of cases classified as mild (early lesions: 19%) or very severe (Hodgkin lymphoma: 14%), and a majority of cases with intermediate/severe histological severity (polymorphic, 21%; monomorphic, 46%). This fact represents a difficulty given that early lesions or lymphomas are precisely the histological types in which there is greater agreement on the type of treatment selected, whereas this is less well established in those cases more frequently diagnosed with polymorphic or monomorphic histological patterns.

According to the results, the histopathological forms of PTLD also appear to be related to the time of PTLD development: severe monomorphic lesions are more frequent in cases of very late onset versus early or late onset (75%, 52% and 34%, respectively). Although there is a progressive reduction in the incidence of PTLD over time, there is a greater risk of developing more histologically severe forms with poorer prognoses, making it necessary to maintain surveillance on PTLD in the long term for all patients.

Although we can consider the response to treatment recorded in the cases included in this series as very good, with a survival rate of 88% of the patients (160/181), it should be noted that one-third developed graft rejection as a consequence of the reduction or cessation of the IS. These results indicate that the general prognosis of PTLD in children with SOT is globally good; however, there is still room for improvement in terms of patient survival as well as in relation to potential complications derived from treatment.

The main limitation of this study in assessing the results of both the survey and the collection of clinical cases is that not all types of SOT are equally represented, which could lead to bias in the interpretation of the results and the conclusions drawn. Another limitation is the potential bias inherent to all online surveys run by physicians.

## 5. Conclusions

In the years since the inception of pediatric transplant programs, a significant improvement has been achieved in reducing the incidence of PTLD and in the results of its treatment. However, it remains one of the most frequent causes of late mortality and morbidity associated with its treatment, with a particular risk of graft loss secondary to rejection, as shown by the results of our series. According to the results obtained in the survey, there are aspects related to the prevention, diagnosis and treatment of PTLD where there is a broad consensus, while there are others where the agreement is less solid, probably due to the lack of clinical evidence in those topics. Multidisciplinary teams are needed to address PTLD prevention, diagnosis and treatment. Identification of patients at higher risk of EBV infection is crucial to define the associated EBV-PTLD risk. There is a lack of international PTLD registry and prospective randomized controlled trials to guide management and therefore clinical trial enrolment should be considered for all patients.

## Figures and Tables

**Table 1 children-08-00661-t001:** Current practices among SOT programs on PTLD treatment.

Tx Program	Heart	I/MV	Kidney	Liver	Lung	Pancreas	Total
(n)	(5)	(5)	(9)	(11)	(2)	(1)	(33)
PTLD Treatment, n (%)							
Decrease IS	3 (60)	4 (80)	5 (56)	5 (45)	2 (100)	1 (100)	20 (61)
Decrease or stop IS	1 (20)	1 (20)	3 (33)	5 (45)	0 (0)	0 (0)	10 (30)
Stop IS	1 (20)	0 (0)	1 (11)	1 (9)	0 (0)	0 (0)	3 (9)
RTX indication (mc), n (%)	4	3	7	8	2	1	25
>2 WHO status	2 (50)	2 (67)	3 (43)	4 (50)	2 (100)	1 (100)	14 (56)
>3 WHO status	3 (75)	2 (67)	3 (43)	5 (62)	2 (100)	1 (100)	16 (64)
EBV levels & clinical	1 (25)	1 (33)	1 (14)	4 (50)	1 (50)	1 (100)	9 (36)
Histological CD20+ B cells	1 (25)	1 (33)	2 (28)	4 (50)	2 (100)	1 (100)	11 (44)
Individual discussion	3 (75)	2 (67)	5 (71)	3 (37)	1 (50)	1 (100)	15 (60)
ND	1	2	2	3	0	0	8
RTX doses, n (%)	4	3	7	10	2	1	27
4 doses, qw	3 (75)	2 (67)	5 (71)	7 (70)	1 (50)	1 (100)	19 (70)
5-6 doses, qw	0 (0)	0 (0)	1 (14)	1 (10)	0 (0)	0 (0)	2 (7)
Other	1 (25)	1 (33)	1 (14)	2 (20)	1 (50)	0 (0)	6 (22)
ND	1	2	2	1	0	0	6
CTLs, n (%)							32
Yes	1 (25)	1 (20)	3 (33)	1 (9)	1 (50)	0 (0)	7 (22)
CTLs indication, n (%)							6
Individual discussion	0 (0)	0 (0)	1 (50)	0 (0)	0 (0)	0	1 (17)
Persistent EBV	1 (100)	1 (100)	1 (50)	1 (100)	1 (100)	0	5 (83)
ND	4	4	7	10	1	1	27
Chemotherapy, n (%)							
Yes	3 (60)	4 (80)	9 (100)	9 (82)	1 (50)	0 (0)	26 (79)
Chemotherapy indications (mc), n (%)	4	4	8	9	2	1	28
Do not respond to RTX	3 (75)	3 (75)	5 (62)	5 (55)	2 (100)	1 (100)	19 (68)
HL type	3 (75)	2 (50)	6 (75)	6 (67)	2 (100)	1 (100)	20 (71)
Monomorphic PTLD	2 (50)	1 (25)	3 (37)	5 (55)	2 (100)	1 (100)	14 (50)
Disease progression after RTX	1 (25)	1 (25)	5 (62)	3 (33)	2 (100)	1 (100)	13 (46)
Other	0 (0)	1 (25)	0 (0)	1 (11)	0 (0)	0 (0)	2 (7)
ND	1	1	1	2	0	0	5
Surgical removal, n (%)							
Yes	3 (60)	3 (60)	5 (56)	6 (55)	1 (50)	1 (100)	19 (58)

Abbreviations: CTLs, cytotoxic T–cell treatment; EBV, Epstein–Barr virus; HL, Hodgkin’s lymphoma; I/MV, intestinal/multivisceral; IS, immunosuppression; mc, multiple choice; ND, no data available; PTLD, post-transplant lymphoproliferative disease; qw, once a week; RTX, rituximab; SOT, solid organ transplantation; Tx, transplantation; WHO, World Health Organization.

**Table 2 children-08-00661-t002:** Cross-sectional cohort of PTLD diagnosed pediatric cases 2012–2016.

Type of Transplant	Heart	I/MV	Kidney	Liver	Lung	Total	*p*-Value
Number of patients, n (%)	3 (2)	13 (10)	17 (14)	90 (71)	3 (2)	126 (100)	ns
Number of surviving, n (%)	3 (100)	9 (69)	15 (88)	81 (90)	3 (100)	111 (88)	ns
Number in full remission, n (%)	3 (100)	8 (61)	15 (88)	73 (81)	3 (100)	102 (81)	ns
Age at Tx (months), median (IQR)	132 (120–192)	84 (48–122.5)	84 (47.5–155.5)	15 (9–30)	120 (60–132)	25 (10–74)	<0.01 ^1^
Age at Dx (months), median (IQR)	204 (168–204)	132 (84.5–168)	157 (102.5–186)	38 (20–64.5)	121 (61–144)	55 (25.5–126.3)	<0.01 ^2^
Time from Tx to PTLD (months), median (IQR)	36 (12–84)	12 (5.5–72)	52 (12–75)	12 (6–35.2)	1 (1–12)	12 (6–42.2)	ns
Early onset, n (%)	0 (0)	4 (31)	2 (12)	40 (44)	2 (67)	48 (38)	
Late onset, n (%)	3 (100)	8 (61)	14 (82)	48 (53)	1 (33)	74 (59)	
Very late onset, n (%)	0 (0)	1 (8)	1 (6)	2 (2)	0 (0)	4 (3)	
WHO classification, n (%)							
1. Non-destructive	0 (0)	0 (0)	0 (0)	23 (25)	0 (0)	23 (18)	
2. Polymorphic	0 (0)	1 (8)	2 (12)	22 (24)	0 (0)	25 (20)	
3. Monomorphic	3 (100)	10 (77)	15 (88)	25 (28)	3 (100)	56 (44)	
4. HL PTLD	0 (0)	2 (15)	0 (0)	18 (29)	0 (0)	20 (16)	
Poly and monomorphic	0 (0)	0 (0)	0 (0)	2 (2)	0 (0)	2 (2)	
EBV viral load (c/mL) ×10^6^, median (range)	3.6 (0.01–7.2)	0.55 (0.06–2.5)	0.05 (0.003–0.8)	0.2 (0.07–1)	ND	0.2 (0.04–1)	ns
Highest LDH (U/L), range	319–700	212–4040	256–49558	168–5110	311–5110	168–49558	ns
Outcome							
Ig supplementation, n (%)	1 (33)	3 (23)	3 (18)	23 (25)	0 (0)	30 (24)	ns
Rejection after remission, n (%)	2 (67)	6 (46)	2 (12)	29 (32)	0 (0)	39 (31)	ns
IS after remission, n/v (%)	1/2 (50)	5/7 (71)	13/13 (100)	59/65 (91)	3 (100)	81/90 (90)	ns
Relapse after remission, n/v (%)	0 (0)	1/2 (50)	0 (0)	5/53 (9)	1 (33)	7/78	ns

^1^ Post hoc tests (using the Bonferroni-Dunn correction to adjust *p*) indicated that median age at liver transplantation was significantly lower than heart Tx (*p* = 0.02), intestinal Tx (*p* = 0.0002) and kidney Tx (*p* ≤ 0.0001). ^2^ Post hoc tests (using the Bonferroni-Dunn correction to adjust *p*) indicated that median age at Dx with PTLD in liver transplantation was significantly lower than heart Tx (*p* = 0.01), intestinal Tx (*p* = 0.0009) and kidney Tx (*p* ≤ 0.0001). Abbreviations: c/mL, copies per milliliter; Dx, diagnosis; EBV, Epstein–Barr virus; HL, Hodgkin´s lymphoma; I/MV, intestinal/multivisceral; IgG, immunoglobulin; IS, immunosuppression; LDH, lactate dehydrogenase; ND, no data available; ns, no significant; PTLD, post-transplant lymphoproliferative disease; Tx, transplantation; U/L, units per litre; n/v, number of cases per valid values; WHO, World Health Organization.

**Table 3 children-08-00661-t003:** Localization of PTLD cases.

Type of Transplant	Heart	I/MV	Kidney	Liver	Lung	Total
PTLD tumour locations, n (%)						
Allograft	0 (0)	2 (15)	2 (12)	9 ^1^ (10)	3 ^2^ (100)	16 (13)
CNS	0 (0)	0 (0)	0 (0)	1 (1)	0 (0)	1 (1)
Disseminated	1 (33)	0 (0)	2 (12)	10 [3] (8)	3 [3] (100)	10 (8)
ENT/cervical LN	0 (0)	5 (38)	3 (18)	35 (39)	0 (0)	43 (34)
LN (other than cervical)	0 (0)	5 (38)	4 (24)	12 [1] (12)	0 (0)	20 (16)
GI tract/abdominal	1 (33)	0 (0)	3 (18)	28 [2] (29)	0 (0)	30 (24)
Bone	0 (0)	0 (0)	2 (12)	0 (0)	0 (0)	2 (2)
BM	1 (0)	1 (33)	0 (0)	1 (1)	0 (0)	3 (2)
Heart	0 (0)	0 (0)	1 (6)	0 (0)	0 (0)	1 (1)

^1^ Six cases presented another localization. They are indicated by square brackets. ^2^ All cases were disseminated and involved the allograft. They are indicated by square brackets. Abbreviations: BM, bone marrow; CNS, central nervous system, ENT: ear–nose–throat; GI, gastrointestinal; I/MV, intestinal/multivisceral; LN, lymph nodes; PTLD, post-transplant lymphoproliferative disease.

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
