# Peer review of "Current Practices on Diagnosis, Prevention and Treatment of Post-Transplant Lymphoproliferative Disorder in Pediatric Patients after Solid Organ Transplantation: Results of ERN TransplantChild Healthcare Working Group Survey"

_children, 2021, doi:10.3390/children8080661_

Round 1

Reviewer 1 Report

The authors present the reuslts of an international working group survey with regard to current practices on several clinical issues in pediatric SOT-related PTLD.

The manuscript is well written and interesting , although I have some comments:

  • Introduction: some recent basic references should be added. I would avoid citing review articles > 5 years ago given the increase in our knowledge on PTLD.
  •  Introduction page 3 second paragraph: the role of alemtuzumab in development of PTLD is very controversial, so I would suggest to omit.
  • Material and methods: page 4: the authors should define which cut off they use for the definition of early onset, late onset and very late onset PTLD.   
  • Results: 2nd paragraph: why do the authors mention the number of transplants during the period 2012-2016; is this information useful?  
  • Results: 3.1.2. this paragraph is the same as the first paragraph of 3.1.1. on IS and PTLD prevention. This should be corrected.
  • Results: 3.1.4. I don't understand the first paragraph: what is the difference between reducing IS versus reducing IS and/or stopping IS and stopping IS. This seems very confusing (but maybe it's logical?).
  • Results: 3.1.4. What do the authors mean by (page 6) 'histological diagnosis > 2 WHO status'?
  • The discussion is a bit too long, this should be shortened. The introduction already provided a nice summary of different apsects on PTLD, so they shouldn't be repeated too much.

Kind regards,

Reviewer 2 Report

The authors present the results of a cross-sectional questionnaire regarding the management of post-transplant lymphoproliferative disease (PTLD) among international centers and between various organ transplant systems in the pediatric population. PTLD represents the most common malignancy in pediatric solid organ transplant recipients. However, there is a lack of consensus on the management.

A total of 34 SOT programs from 13 European 190 centres and 9 countries participated in the study, which is remarkable. Interestingly, up to 62% of the centres the decision to start pre-emptive treatment for PTLD is based on EBV viraemia plus additional methods. However, there is not sufficient evidence or general consensus in using any of the additional methods. Overall, there were similar approaches to 1) initial strategy for the management of PTLD, entailing the reduction of immunosuppression, 2) diagnostic evaluation of PTLD, including mostly imaging, and 3) treatment of PTLD, including the use of rituximab and chemotherapy among the centres.

As reported previously, mortality was lower compared to adult SOT. However, more than 30% of patients developed graft rejection.

Overall, the authors were able to identify aspects of the management of PTLD in the pediatric SOT recipients where there is less consensus among different centres and will clearly benefit from further evaluations.

Minor

  • Page 5, rows 240-248. The text is exactly as in page 4, rows 199-204. It is redundant. In my opinion, the text in page 5, rows 240-248 can be deleted.
  • Section 3.1.2 and section 3.1.3 have the same title PTLD diagnostic and evaluation tests. In my opinion, section 3.1.2 can be deleted
  • Table 2 needs to be formatted. In the Time from Tx to PTLD section: early-onset, late-onset, and very late-onset are not aligned with n (%).

Round 2

Reviewer 1 Report

One small remark: page 7 line 3: "histological diagnosis (pathologic category) >2 WHO status": I would replace this by "monomorphic subtype" as it remains a bit confusing (it's not a categorical variable) to me.

Author Response

Dear reviewer,

Thank you for your comments. Please find below my response.

Best regards,

Yadira

Point 1: One small remark: page 7 line 3: "histological diagnosis (pathologic category) >2 WHO status": I would replace this by "monomorphic subtype" as it remains a bit confusing (it's not a categorical variable) to me.

Response 1: Thank you very much, I have changed it for: "Histological diagnosis (pathologic category) monomorphic subtype"